# Data-Oriented Constitutive Modeling of Plasticity in Metals

**DOI:** 10.3390/ma13071600

**Published:** 2020-04-01

**Authors:** Alexander Hartmaier

**Affiliations:** ICAMS, Ruhr-Universität Bochum, 44801 Bochum, Germany; alexander.hartmaier@rub.de

**Keywords:** plasticity, machine learning, constitutive modeling

## Abstract

Constitutive models for plastic deformation of metals are typically based on flow rules determining the transition from elastic to plastic response of a material as function of the applied mechanical load. These flow rules are commonly formulated as a yield function, based on the equivalent stress and the yield strength of the material, and its derivatives. In this work, a novel mathematical formulation is developed that allows the efficient use of machine learning algorithms describing the elastic-plastic deformation of a solid under arbitrary mechanical loads and that can replace the standard yield functions with more flexible algorithms. By exploiting basic physical principles of elastic-plastic deformation, the dimensionality of the problem is reduced without loss of generality. The data-oriented approach inherently offers a great flexibility to handle different kinds of material anisotropy without the need for explicitly calculating a large number of model parameters. The applicability of this formulation in finite element analysis is demonstrated, and the results are compared to formulations based on Hill-like anisotropic plasticity as reference model. In future applications, the machine learning algorithm can be trained by hybrid experimental and numerical data, as for example obtained from fundamental micromechanical simulations based on crystal plasticity models. In this way, data-oriented constitutive modeling will also provide a new way to homogenize numerical results in a scale-bridging approach.

## 1. Introduction

Finite element analysis (FEA) is a widespread numerical tool for studying the mechanical behavior of structures. While in many applications it is sufficient to know under which conditions a part of the structure fails plastically or suffers damage or fracture, in some cases, like in sheet forming or for crash simulations, it is important to be able to simulate the plastic deformation during loading and to obtain the shape of the structure after the external load is released. Such non-linear behavior is typically described by constitutive models that relate stress and strain in a material, as described in any textbook on non-linear finite element modeling, e.g., see [1]. Conventionally, constitutive relations for plasticity are formulated as flow rules based on a plastic potential. In the simplest case, the latter is the yield function of the material, determining at which local stress the material starts yielding plastically and which plastic strain increment will result from such plastic deformation. As described in the next section, such yield functions relate the equivalent stress and the yield strength of a material, which needs to be determined experimentally or with the help of more fundamental models in a scale-bridging approach.

Experimentally, the yield strength is typically determined in uniaxial tensile tests, in which materials frequently exhibit an anisotropy in their plastic behavior, i.e., the yield strength depends on the orientation of the loading axis with respect to the material coordinate system, defined for example by rolling, normal, and transverse direction. In conventional approaches such anisotropic plastic behavior is described by defining a proper equivalent stress that takes into account material anisotropy, such that yielding occurs at a constant scalar yield strength. This approach has been introduced by Hill [2] and applied to orthotropic plasticity in sheet metals [3]. The concept has been generalized to linear transformation-based anisotropic yield functions by Barlat et al. [4] and to methods describing distortions of the yield surface caused by anisotropic work hardening [5]. In forming technology, similar ideas have been successfully applied to predict the resulting shape of sheet metals after deep drawing [6]. All these approaches have in common that the information about the material anisotropy is mapped into the definition of the equivalent stress, while they differ in the amount of material parameters that is required to describe the anisotropy in the material’s flow behavior. To determine these parameters, a series of experiments with different mutual orientations of loading axis and material axis is necessary. Alternatively, micromechanical models, in which discrete representations of the material’s microstructure are used together with quite fundamental crystal plasticity models, can be used to calculate the anisotropic flow behavior of a polycrystalline metal [7,8].

To model non-linear material behavior, in more recent approaches, the method of data-based mechanics has been introduced by Kirchdoerfer and Ortiz [9], in which stress-strain data from experimental tests are used directly, rather than using constitutive rules. While the first approaches have been limited to elastic structures under static loads, recently, this concept has been extended to dynamics [10] and to inelastic material behavior [11]. These methods are based on a fundamental re-formulation of the basic equations of mechanics and thus require completely new mechanical solvers. Other data-driven methods in plasticity are formulated as process models, e.g., for air-bending [12], or focus on the application of data-oriented methods as constitutive models in computational plasticity [13]. The latter idea allows the use of existing FEA solvers for mechanical problems, and is also followed in this work, in which a new formulation of a data-oriented flow rule is introduced that can replace conventional constitutive models—formulated in a mathematical closed form—by machine learning (ML) algorithms. In this data-oriented formulation, the anisotropy of the material’s flow behavior is considered in a directionally dependent yield strength of the material rather than by an anisotropic transformation of the stress. Using ML algorithms as yield functions provides a great flexibility to describe arbitrary mathematical functions, and at the same time, holds the potential to handle large data sets and multi-dimensional feature vectors as input. Hence, using ML algorithms as constitutive rules for plastic material behavior offers the possibility to explicitly take into account the microstructural information of the material in constitutive modeling. Furthermore, data resulting from experiment and micromechanical simulations can be hybridized to generate training data sets. An overview on applications of ML and data-mining methods in continuum mechanical simulations of material behavior has been provided by Bock et al. [14].

ML algorithms can be classified into algorithms for supervised and unsupervised learning. The former group can be further categorized into classification algorithms, which divide a multi-dimensional feature space into regions with similar properties, and into regressors, which provide linear or non-linear regression functions for the given multi-dimensional data set. Support Vector Machines (SVM) are successfully applied both as classifiers (SVC) [15] and regressors (SVR) [16,17]; an overview on both applications is given in a technical report by Gunn [18]. Since yield functions in continuum plasticity are also employed to subdivide stress space into elastic and plastic regions, this work aims at investigating the possibility of using SVC for the purpose of constitutive modeling in plasticity. An overview on data-mining methods and statistical learning, also covering the SVM method, is given by Hastie et al. [19].

The present paper is organized as follows: In the next section, the basic concepts of continuum plasticity are briefly summarized, and a new mathematical formulation is introduced, which enables a data-oriented approach to constitutive modeling. Subsequently, a consistent formulation of the SVC method to serve as yield function in continuum plasticity is introduced, which is then trained with artificial data resulting from a Hill-like reference material. Using such data has the advantage of being able to judge the quality of the approximations in an objective way. In the next step, trained ML yield function is applied as constitutive model in simple finite element simulations (Appendix A) to demonstrate its applicability for this purpose. Finally, the results of ML and conventional flow rules are compared and the conclusions drawn from this comparison are presented.

## 2. Methods

### 2.1. Anisotropic Continuum Plasticity

In order to describe the elastic-plastic deformation of a material, we introduce the strain tensor ϵ that describes the deformation of the material and the stress tensor σ that describes the forces acting on the surface of the material. Note that tensorial quantities with rank ≥1 are typeset in bold letters, whereas scalar quantities are represented by standard characters. In the elastic regime, Hooke’s law is used as constitutive relation between stress and strain, such that
(1)σ=Cϵ,
where C is the fourth-rank elasticity tensor of the material. To describe plastic deformation, the yield function of the material is introduced as
(2)$$f\left(\sigma \right)=\sigma_{\mathrm{eq}}-\sigma_{y},$$
which takes negative values if the equivalent stress $\sigma_{\mathrm{eq}}$ is smaller than the yield strength $\sigma_{y}$ of the material, i.e., in the elastic regime. When f=0 plastic yielding sets in, and in case of work hardening, $\sigma_{y}$ should be considered as flow stress after this point. Since this work only deals with the onset of plastic yielding, ideal plasticity will be assumed throughout, such that $\sigma_{y}$ is a constant, irrespective of the deformation history of the material. Denoting the principal stresses of the stress tensor σ as $\sigma_{j}$ with (j=1,2,3), the equivalent stress takes the form
(3)$$\sigma_{eq}^{J2}=\sqrt{\frac{1}{2}\left[{\left(\sigma_{1}-\sigma_{2}\right)}^{2}+{\left(\sigma_{2}-\sigma_{3}\right)}^{2}+{\left(\sigma_{3}-\sigma_{1}\right)}^{2}\right]},$$
which—following the definition of von Mises (see, e.g., the translation of the original work by D. H. Delphenich [20])—is based on the second invariant of the stress deviator (J2). In conjunction with the yield function of Equation (Equation 2), it describes the onset of plastic yielding for isotropic materials. Note that the formulation in Equation (Equation 3) is intrinsically independent of hydrostatic stress components p=1/3Trace(σ) and thus does not require to explicitly calculate the deviatoric stress
(4)$$\sigma^{\prime }=\sigma -pU,$$
where U is the unit tensor. By this definition of the equivalent stress, it is inherently assumed that hydrostatic stress components do not affect the plastic flow behavior of the considered material, which is typically fulfilled for metals, but not for polymers or rocks, such that the method formulated here, will mainly apply to metallic materials or, more generally, to materials, where hydrostatic stresses do not influence the plastic behavior.

As described in the introduction, many materials exhibit a directionally dependent yield strength, such that anisotropic flow rules need to be introduced. A first definition of such anisotropic flow rules was introduced by Hill [2], who used a generalized definition of the equivalent stress to achieve a directionally dependent mapping of the equivalent stresses to maintain a constant yield strength. Hence, in this formulation, the anisotropy is considered in the stress rather than in the yield strength. Since the mathematical formulations in this work are purely based on principal stresses, we use a simplified version of the Hill definition and introduce a Hill-like anisotropic definition of the equivalent stress as
(5)$$\sigma_{\mathrm{eq}}=\sqrt{\frac{1}{2}\left[H_{1}{\left(\sigma_{1}-\sigma_{2}\right)}^{2}+H_{2}{\left(\sigma_{2}-\sigma_{3}\right)}^{2}+H_{3}{\left(\sigma_{3}-\sigma_{1}\right)}^{2}\right]},$$
with only three material parameters $H_{1},H_{2}$ and $H_{3}$, whereas in his original work, Hill introduced three more parameters for an orthotropic material to scale also the shear stress components. Since for orthotropic materials loaded along the main material axes there is no mutual influence of shear and normal components of stress and strain, the formulation introduced here is restricted to loading situations that only produce normal stresses and strains, and where consequently all off-diagonal components of stress and strains tensors remain zero. Furthermore, it is assumed that the loading axes and the main axes of the orthotropic material coincide. Hence, this formalism is currently only valid for a small subset of loading conditions for materials with orthotropic flow anisotropy. The definition of the equivalent stress following Hill can be considered as a generalization of the J2 equivalent stress, because for isotropy, i.e., $H_{1}=H_{2}=H_{3}=1$, both definitions are equal.

The restrictions applied in this work, allow the mathematical notation to be simplified by only considering principal stresses. In future work, it is intended to render the formulation more general by exploiting that for any stress state, there exists a coordinate system in which the given stress tensor becomes a diagonal tensor composed of the principal stresses $\sigma_{j}$. This coordinate system is given by the eigenvectors of the stress, representing the principal directions, such that the coordinate system of the original stress tensor—and with it the material axes—can be rotated into the coordinate system of the eigenvectors of the stress tensor. In this orientation the stress tensor becomes a diagonal tensor, and Equation (Equation 5) can be evaluated with parameters $H_{i}^{\prime }$ in the rotated state of the material axes.

The thus defined yield function can be used to determine whether a given stress state results in a purely elastic or rather in an elastic-plastic deformation of a material. The condition f(σ)=0 relates stresses lying on a specific hyperplane in stress space, the so-called the yield-locus. Since a material does not sustain any stresses larger than the yield stress (for ideal plasticity) or the flow stress (in case of work hardening), acceptable stress states either produce a negative value of the yield function (elasticity) or lie on the yield locus (plasticity), which should be a convex hull of the elastic stress states. Hence, if a predictor step in finite element analysis (FEA) produces a stress outside the yield locus, a plastic strain increment must be calculated that leads again to an accepted stress state on the yield locus. The return mapping algorithm to calculate such strain increments has been described in many text books on continuum plasticity and non-linear FEA, such that here only a very brief summary based on [1] is reproduced. According to the Prandtl–Reuss flow rule, the plastic strain increment for a given time step can be calculated as
(6)$${\dot{\epsilon }}_{p}=\dot{\lambda }\frac{\partial f}{\partial \sigma }=\dot{\lambda }n,$$
where n is the normal vector to the yield locus, defined by the gradient of the yield function ∂f/∂σ, and $\dot{\lambda }>0$ is the so-called plastic strain multiplier that can be evaluated as
(7)$$\dot{\lambda }=\frac{n\cdot C\dot{\epsilon }}{n\cdot Cn},$$
where $\dot{\epsilon }$ is the total strain increment of the FEA predictor step that leads to a stress state outside the yield locus and which is consequently decomposed into the plastic strain increment, given by Equation (Equation 6), and the elastic strain increment or stress increment given by
(8)$$\dot{\sigma }=C_{t}\dot{\epsilon }$$
with the tangent stiffness tensor
(9)$$C_{t}=C-\frac{Cn⊗Cn}{n\cdot Cn}$$
where “⊗” denotes the tensorial product in the form $a_{i}⊗b_{j}=a_{i}b_{j}$.

The gradient of the yield function with respect to the principal stresses can be evaluated analytically as
(10)$$\begin{array}{ccc} \frac{\partial f}{\partial \sigma_{1}}=\frac{\partial \sigma_{eq}}{\partial \sigma_{1}} & = & \frac{\left(H_{1}+H_{3}\right)\sigma_{1}-H_{1}\sigma_{2}-H_{3}\sigma_{3}}{\sigma_{\mathrm{eq}}}\\ \frac{\partial f}{\partial \sigma_{2}}=\frac{\partial \sigma_{eq}}{\partial \sigma_{2}} & = & \frac{\left(H_{2}+H_{1}\right)\sigma_{2}-H_{1}\sigma_{1}-H_{2}\sigma_{3}}{\sigma_{\mathrm{eq}}}\\ \frac{\partial f}{\partial \sigma_{3}}=\frac{\partial \sigma_{eq}}{\partial \sigma_{3}} & = & \frac{\left(H_{3}+H_{2}\right)\sigma_{3}-H_{3}\sigma_{1}-H_{2}\sigma_{2}}{\sigma_{\mathrm{eq}}} \end{array}$$

Note that in the case of isotropic plasticity ($H_{1}=H_{2}=H_{3}=1$), the gradient takes the simple form
(11)$$\frac{\partial f}{\partial \sigma }=3\frac{\sigma -pU}{\sigma_{\mathrm{eq}}}=3\frac{\sigma^{\prime }}{\sigma_{\mathrm{eq}}}.$$

This section served the purpose to introduce the main physical quantities in the notation used in this work. For further details of continuum plasticity or FEA, the reader is referred to standard textbooks, as for example [1]. In the following, the formalism for the data-oriented constitutive model based on a machine learning (ML) yield function is laid out.

### 2.2. Stress Space in Cylindrical Coordinates

Since plastic deformation in most metals does not depend on hydrostatic stress components, it is useful to transform principal stresses from their representation as a 3-dimensional (3D) Cartesian vector of principal stresses $\sigma =(\sigma_{1},\sigma_{2},\sigma_{3})$ into a cylindrical coordinate system with $s=(\sigma_{\mathrm{eq}},\theta ,p)$, where the equivalent stress $\sigma_{\mathrm{eq}}$ represents the norm of the stress deviator $\sigma^{\prime }$, and the polar angle θ lies in the deviatoric plane normal to the hydrostatic axis *p*, which has already been used by Hill [2]. This coordinate transformation improves the efficiency of the training, because only two-dimensional data for the equivalent stress and the polar angle need to be used as training features, whereas the hydrostatic component is disregarded. Hence, by exploiting basic physical principles, we effectively reduce the dimensionality of the problem from 6 independent components of an arbitrary stress tensor to 2 degrees of freedom, without loosing the generality of the formulation. As the polar angle θ can be considered a generalized Lode angle [21], it is noted that the Lode angle, by definition, describes the axiality of a loading state in a way that uniaxial loads in different directions result in the same Lode angle. Since our formulation aims at describing anisotropy in the plastic deformation, uniaxial stresses in different directions must possess different angles. To achieve this, we introduce a complex-valued deviatoric stress
(12)$$\sigma_{c}^{\prime }=\sigma \cdot a+i\sigma \cdot b=\sqrt{2/3}\sigma_{\mathrm{eq}}e^{i\theta },$$
where *i* is the imaginary unit, such that the polar angle
(13)$$\theta =\arg\sigma_{c}^{\prime }=-i\;\ln\frac{\sigma \cdot a+i\sigma \cdot b}{\sqrt{2/3}\sigma_{\mathrm{eq}}}.$$
with the unit vectors $a=\left(2,-1,-1\right)/\sqrt{6}$ and $b=\left(0,1,-1\right)/\sqrt{2}$ that span the plane normal to the hydrostatic axis $c=\left(1,1,1\right)/\sqrt{3}$.

To transform the gradient of the yield function from this cylindrical stress space back to the principle stress space, in which form it is used to calculate the direction of the plastic strain increments in the return mapping algorithm of the plasticity model, we introduce the Jacobian matrix for this coordinate transformation as
(14)$$J=\frac{\partial s}{\partial \sigma }=\left(\begin{array}{ccc} \frac{\partial \sigma_{\mathrm{eq}}}{\partial \sigma_{1}} & \frac{\partial \theta }{\partial \sigma_{1}} & \frac{\partial p}{\partial \sigma_{1}}\\ \frac{\partial \sigma_{\mathrm{eq}}}{\partial \sigma_{2}} & \frac{\partial \theta }{\partial \sigma_{2}} & \frac{\partial p}{\partial \sigma_{1}}\\ \frac{\partial \sigma_{\mathrm{eq}}}{\partial \sigma_{3}} & \frac{\partial \theta }{\partial \sigma_{3}} & \frac{\partial p}{\partial \sigma_{1}} \end{array}\right)$$
where $\partial \sigma_{\mathrm{eq}}/\partial \sigma_{j}$ is given in Equation (Equation 10), $\partial p/\partial \sigma_{j}=1/3$ and
(15)$$\frac{\partial \theta }{\partial \sigma }=-i\;\left(\frac{a+ib}{\sigma \cdot a+i\sigma \cdot b}-\frac{3\sigma^{\prime }}{\sigma_{\mathrm{eq}}^{2}}\right).$$

With this Jacobian, the gradient can be calculated as
(16)$$\frac{\partial f}{\partial \sigma }=J\frac{\partial f}{\partial s}.$$

### 2.3. Data-Oriented Yield Function

While the concept of mapping the equivalent stress in describing anisotropic flow behavior has been applied successfully in the approaches of Hill [2,3] and Barlat [4], for a data-oriented yield function, it is impracticable to calculate the necessary parameters for this stress mapping explicitly. Hence, it is of advantage to reformulate the flow rule in such a way that the yield strength is considered to be directionally dependent, whereas the equivalent stress is formulated in an objective way, without prior knowledge of the material behavior. This is achieved by using the J2 equivalent stress in the flow rule and formally considering the flow stress to be a function of the polar angle in the deviatoric plane, such that
(17)$$f_{d}\left(s\right)=s_{1}-\sigma_{y}\left(s_{2}\right)=\sigma_{eq}^{J2}-\sigma_{y}\left(\theta \right).$$

A further advantage of this formulation is that the dependence of the yield function on the two degrees of freedom of the cylindrical stress notation is separated into two independent terms. Furthermore, for symmetry reasons, it is required that the yield strength is a periodic function of the polar angle with periodicity 2π. The gradient of the ML yield function w.r.t. the cylindrical coordinates reads
$$\begin{array}{ccc} \frac{\partial f_{d}}{\partial s_{1}} & = & \frac{\partial f_{d}}{\partial \sigma_{eq}^{J}2}=1\\ \frac{\partial f_{d}}{\partial s_{2}} & = & \frac{\partial f_{d}}{\partial \theta }=\frac{d\sigma_{y}}{d\theta }\\ \frac{\partial f_{d}}{\partial s_{3}} & = & \frac{\partial f_{d}}{\partial p}=0. \end{array}$$

It is seen that in the cylindrical stress space $\partial f_{d}/\partial \sigma_{\mathrm{eq}}=1$ and $\partial f_{d}/\partial p=0$, under the condition that plasticity is independent of hydrostatic stress components. Hence, the only non-constant component of the gradient is $\partial f_{d}/\partial \theta$, which simplifies the numerical implementation of the method. For isotropic J2 plasticity, $\partial f_{d}/\partial \theta =0$, and in this case it is particularly easy to calculate the gradient and to see that the formulations in both coordinate systems result in the same gradient. The transformation of this gradient into the principal stress space is achieved by multiplication with the Jacobian, according to Equation (Equation 16).

To establish a data-oriented formulation, we introduce a yield function in the form of a machine learning (ML) algorithm, rather than in a mathematically closed form with a number of model parameters that need to be fitted to the data. This enables us to use the available data directly for the training of the ML algorithm. Furthermore, ML methods allow for the use of higher dimensional feature vectors such that in future work, information about the material properties and the microstructure of the material can be directly used as input into one single ML yield function able to handle different microstructures.

In this work, Support Vector Classification (SVC) is applied to categorize data sets consisting of principal stresses into the classes “elastic” and “plastic”. During training, SVC constructs a hyperplane in stress space, which separates the two regions from each other. Consequently, this hyperplane, defined by the zeros of the so-called SVC decision function, is the equivalent to the yield locus, defined by the zeros of the yield function, and it is constructed such that it has the largest distance to the nearest training data points of both classes. The SVC decision function is defined as [15]
(18)$$f_{\mathrm{SVC}}\left(s\right)=\sum_{k=1}^{n}y_{k}\alpha_{k}K\left(s_{\mathrm{sv}}^{\left(k\right)},s\right)+\rho ,$$
where *n* is the number of support vectors identified during the training process and
(19)$$K\left(s_{\mathrm{sv}}^{\left(k\right)},s\right)=\exp\left(-\gamma ∥s-s_{\mathrm{sv}}^{\left(k\right)}{∥}^{2}\right)$$
is the radial basis function (RBF) kernel of the SVC, which is well suited for non-linear problems, with the parameter γ that determines how fast the influence of one support vector decays in stress space. The support vectors $s_{\mathrm{sv}}^{\left(k\right)}$, the dual coefficients $y_{k}\alpha_{k}$, and the intercept ρ are determined during the training. There are essentially two parameters that control the training process and thus the quality of the obtained decision function: (i) γ>0 is a parameter of the RBF kernel function and controls how far-reaching the influence of each support vector is: the larger the value of γ, the more short-ranged and local the influence; (ii) C>0 is a parameter that is used only during the training to regularize the decision function, but that does not directly enter the decision function (Equation 18). The larger the value of *C*, the more flexible but irregular the decision function will become by approximating the shape of the training data more accurately. The choice if these training parameters is critical for the successful use of the decision function in a flow rule. In short, the larger both values are, the more flexible and sensitive to local values the resulting decision function will become. Thus, too small values will result in a smooth but not accurate approximation of the true yield function, whereas too large values will result in a noisy yield function that cannot be used in FEA. The numerical example presented later on will demonstrate this effect and provide examples for values producing accurate yet sufficiently smooth results for the yield function.

For the supervised training, a set of $n_{t}$ feature vectors $s_{\mathrm{train}}^{\left(j\right)}$ has to be provided together with the result vector $y^{\left(j\right)}$ with $j=(1,\ldots ,n_{t})$, which takes values only in two categories: $y^{\left(j\right)}=-1$ for those training data points $s_{\mathrm{train}}^{\left(j\right)}$ in the “elastic” regime and $y^{\left(j\right)}=+1$ for training data in the “plastic” regime. This training data are within the core of the method outlined here, because during the training process they are directly used to define the support vectors that in turn determine the plastic properties of the material. It is, therefore, essential to have sufficiently many training data points in close proximity to the yield locus to approximate it accurately. However, the SVC training will create support vectors only in the region covered by the training data, and outside this region the decision function drops to zero, which could produce erroneous results if the elastic predictor step of the return mapping algorithm falls into such a region. Hence, data points that lie deeper within the elastic and plastic regions are required to prevent the decision function (Equation 18) from falling back to zero. Such data points, however, can be constructed from available raw data lying close to the yield locus simply by linearly scaling principal stresses in the elastic region towards smaller values, such that they stay within the elastic region, and, likewise in the plastic region, scaling principal stress data towards higher values. Thus, the raw data can be spread throughout the stress space, even without knowing the strain value associated with each data point. Only the knowledge of its class “elastic” or “plastic” is required as knowledge for this data extension step improving the training process. This property of the formalism introduced in this work is of critical importance, because it enables the creation of large volumes of training data from relatively few raw data points close to the yield locus, as demonstrated below.

As seen from Equation (Equation 18), the decision function is a continuous function constructed in a way to reproduce this category, i.e., the sign of the training data in the respective regions in the optimal way. To make predictions about the elastic or plastic material behavior at any given stress, the sign of the value of the decision function $f_{\mathrm{SVC}}\left(s\right)$ at the given stress is evaluated. Furthermore, the yield locus of the material can be obtained in the same way as for traditional yield functions, simply by finding the zeros of the continuous function. In this way, furthermore, the distance of any point in stress space to the yield locus can be evaluated, which is important for making efficient predictor steps during FEA and for calculating plastic strain increments for the return mapping algorithm. In particular, as described in the previous section, the gradient to the yield locus needs to be known in order to calculate plastic strain increments, that bring the stress back to the yield surface, because the plastic material does not support any stresses outside. Due to the definition of the ML yield function as convolution sum over the support vectors, the gradient to the SVC decision function can be calculated analytically as
(20)$$\frac{\partial f_{\mathrm{SVC}}}{\partial s}=\sum_{k=1}^{n}y_{k}\alpha_{k}\frac{\partial K(s_{\mathrm{sv}}^{\left(k\right)},s)}{\partial s}$$
with
(21)$$\frac{\partial K(s_{sv}^{\left(k\right)},s)}{\partial s}=-2\gamma \exp\left(-\gamma ∥s-s_{\mathrm{sv}}^{\left(k\right)}{∥}^{2}\right)\left(s-s_{\mathrm{sv}}^{\left(k\right)}\right).$$

The gradient of the yield function in the 3D principle stress space is obtained by multiplication of the gradient in the cylindrical stress space with the Jacobian defined in Equation (Equation 14). Thus, the formulation of the data-oriented yield function based on the SVC algorithm can be used directly as ML yield function in FEA, with the same formalism for plasticity as for standard yield functions.

## 3. Results

In the following, it will be demonstrated how the derived formulation of the ML yield function can be trained with data obtained from a reference material and used in FEA as constitutive law for plasticity. All numerical examples are conducted with the tools provided on the open-source platform Sci-Kit Learn [22] and a Python library for FEA written by the author of this work. The Python code used for generating the results presented here is provided as Appendix A in form of a Jupyter notebook.

### 3.1. Training of ML Yield Function

A reference material with Hill-type anisotropy is defined with the material parameters as given in Table 1. The yield locus of this reference material for plane-stress conditions with $\sigma_{3}=0$ is plotted in Figure 1, in which the yield locus of the reference material with Hill-like anisotropy is compared with that of an isotropic material with the same yield strength $\sigma_{y}$.

The thus-defined reference material is used to produce training data—and later also test data—for the machine learning algorithm. To accomplish this, a set of stress values in form of principal deviatoric stresses is produced in a way to cover the complete space of polar angles and also sufficiently many equivalent stresses in the elastic and plastic regimes. This is conveniently achieved by creating a set of $n_{\mathrm{ang}}$ equally distributed polar angles $\theta^{\left(k\right)}$ in the range of [−π,π] and a set of $n_{s}$ equivalent stresses $\sigma_{eq}^{J2(l)}$ in the range $\left[0.1\sigma_{y},5\sigma_{y}\right]$. Note that for the entire procedure, the yield strength $\sigma_{y}$ of the material is assumed to be known. This is not a restriction, because the yield strength of an unknown material can be easily determined from the input data in a pre-analysis step.

The transformation into principal stresses is performed as
(22)$$\sigma_{\mathrm{train}}^{\left(j\right)}=\sqrt{2/3}\sigma_{eq}^{J2(l)}\;\left(a\cos\theta^{\left(k\right)}+b\sin\theta^{\left(k\right)}\right)$$
with
(23)$$j=k+\left(l-1\right)n_{\mathrm{ang}}\;\;\left(k=1,\ldots ,n_{\mathrm{ang}};\;l=1,\ldots ,n_{s}\right)$$
and the unit angles a and b spanning the deviatoric stress plane as given above. This produces a set of $n_{t}=n_{\mathrm{ang}}n_{s}$ principal stresses with which, finally, the set of result vectors
(24)$$y^{\left(j\right)}=\mathrm{sgn}\left(f(\sigma_{\mathrm{train}}^{\left(j\right)})\right)\;\;\mathrm{with}\;\;j=\left(1,\ldots ,n_{t}\right),$$
is generated by evaluating the yield function *f* of the reference material, as defined in Equations (Equation 2) and (Equation 5), with the material parameters given in Table 1.

In the numerical example given here, the full training data set comprises $n_{\mathrm{ang}}=36$ values for the polar angle and $n_{s}=28$ values for the equivalent stress for each angle, resulting in a total of $n_{t}=1008$ training data sets. Concerning the effort to create this data, it is noted here that only the number of angles $n_{\mathrm{ang}}$ is relevant for the number of experiments or micromechanical simulations necessary to generate the training data, because each angle defines a load case from which several stresses in the elastic and plastic regime will result, and other training data points can be easily constructed from this raw data by linear scaling, as described above. The implications of the number of load cases required to achieve an accurate representation of the ML yield function will be further discussed in Section 4. A graphical representation of the training data is shown in Figure 2, where also the different ways of representing the anisotropy of the yield function with Hill-type equivalent stresses and von Mises (J2) equivalent stresses are demonstrated. The actual training data comprise four additional sets of polar angles associated with larger equivalent stresses scaled to values of up to $\sigma_{\mathrm{eq}}=5\sigma_{y}$ to prevent the decision function from falling back to zero in this regime, which might cause erroneous results in FEA. To enforce the periodicity of the training ML yield function and its gradient, the training data is periodically repeated within the training algorithm, such that the polar angle covers a range −1.3π<θ<1.3π.

With this data set, the training of the SVC algorithm is performed. Using the training parameters C=10 and γ=4 results in a very good training score of above 99%. However, to evaluate the true quality of the training procedure and to judge whether overfitting has occurred, it is necessary to verify the results with an independent set of test data, which has not been used for training purposes. The error produced on such test data sets with 480 random deviatoric stresses as data points is below 1%, and the $R^{2}$-correlation coefficient between test data and training data is above 98%, which leads to the conclusion that the trained ML yield function has a very high accuracy and robustness. A variation of the training parameters revealed that the results are rather insensitive to the parameter *C*, which can be varied between 2<C<20 without having a pronounced influence on the results, whereas changing the parameter γ by more then 20% causes a significant deterioration of the training results. The resulting SVC decision function, defined in Equation (Equation 18), is plotted together with the training data in Figure 3 in the deviatoric stress space.

Finally, to demonstrate the quality of the ML yield function in the full principal stress space, the predicted categories are plotted in three different slices corresponding to different plane-stress conditions, together with the yield function of the reference material in Figure 4. It is seen that the training data in the deviatoric space covers in fact only a single line in each slice, which demonstrates the power of reducing the dimensionality of the data-oriented yield function by exploiting basic physical principles.

### 3.2. Application of The Trained ML Yield Function in FE Analysis

The ML yield function trained and analyzed in the previous step shall now be applied in FEA to demonstrate its usefulness for this purpose. The numerical examples provided here have been conducted with the Python library “pyLab-FE” created by the author, which is provided in the Appendix A together with a Jupyter notebook following the work-flow defined in this work. The known parameters and support vectors resulting from the training process of the ML yield function together with the mathematical formalism laid out in Section 2 allow a rather straightforward evaluation of the yield function as sum over the support vectors convoluted with the kernel function, such that they can also be used for implementing a user material subroutine (UMAT) for common commercial FEA tools in any compiler language.

As numerical examples, four different load cases are simulated with FEA: (i) uniaxial stress in horizontal direction, (ii) uniaxial stress in vertical direction, (iii) equibiaxial strain under plane-stress conditions, and (iv) pure shear strain under plane-stress conditions. The simple finite element model used to study these load cases consists of four quadrilateral elements with linear shape functions and full integration, as shown in Figure 5. For all load cases, plane stress conditions with $\sigma_{3}=0$ are enforced and the normal degrees of freedom (dof) for the boundary nodes are prescribed, while all boundary nodes are allowed to relax to their equilibrium positions along the boundary. The bottom and the left-hand-side nodes are always restricted to a normal displacement of zero. For the uniaxial load cases, tensile displacements are prescribed either on the top or on the right-hand-side (rhs) nodes, while the other boundary is force-free, resulting in a uniaxial stress. For equibiaxial strain, the top and the rhs boundary nodes are subjected to identical displacements; whereas for pure shear, the rhs nodes are loaded with the negative displacement applied on the top nodes. By virtue of these boundary conditions, the deformation causes only normal stresses and strains, but no shear components. Hence, the restrictions of the formulation of the ML flow rule are fulfilled, and the material axes remain aligned with the directions of the principal stresses.

These four load cases are applied to the reference material as well as to the material with the ML yield function, and the resulting yield stresses and plastic strains at the end of each load step are compared in Table 2. In Figure 6, the resulting global equivalent stresses and equivalent total strains
(25)$$\epsilon_{\mathrm{eq}}=\sqrt{\epsilon :\epsilon }$$
for each load case are plotted for both materials, where the different definitions of the equivalent stress have been applied to the reference material.

To further demonstrate the correctness of the plastic behavior resulting from the ML flow rule, the flow stresses of the material, i.e., the stresses occurring during plastic deformation, are plotted together with the yield locus. For ideal plasticity, treated in this work, it is expected that the flow stress remains on the yield locus, since the material does not sustain larger stresses. In Figure 7 it is shown that this expectation is fulfilled to a very good degree, by comparing the yield loci and the element solutions of the flow stresses obtained for the ML yield function with those of the reference material with a Hill-like flow rule.

### 3.3. Tresca Flow Rule

In the next example, the ML yield function is trained with data from a reference material with a Tresca yield function, which is fundamentally different from the elliptical Hill-like yield functions. The Tresca equivalent stress is defined as
(26)$$\sigma_{eq}^{\mathrm{Tresca}}=\sigma_{I}-\sigma_{\mathrm{III}},$$
where $\sigma_{I}$ is the largest principle stress and $\sigma_{\mathrm{III}}$ is the smallest principle stress [23]. Using the Tresca equivalent stress in the yield function (Equation 2) leads to an isotropic plastic deformation of the material, however, with very different characteristics than for a J2 equivalent stress. Hence, it is a critical test for the new method developed in this work to apply it to such yield functions. The training data for this yield function has been produced in the same way as before. However, with $n_{\mathrm{ang}}=600$ data points for the polar angle, a much larger number of training data points has been required to follow the subtle features of the Tresca flow rule. Furthermore, the training parameters C=50 and γ=9 have been applied to allow for sufficient flexibility of the ML yield function to approximate the abrupt changes of the yield behavior in the deviatoric stress space, *cf.*
Figure 8. The scores with test and training data are above 99%, and the $R^{2}$-value on test data is 96%. Even with this training procedure, the sharp corners of the Tresca yield locus are slightly rounded off by the ML yield function, leading to somewhat lower yield strengths of the material in these directions, as seen in Figure 9a, where the resulting stress-strain curves are plotted as J2 equivalent stress vs. equivalent total strain. In Figure 9b the yield loci of the Tresca reference material, the trained ML yield function and an isotropic J2 material are plotted in comparison. Furthermore, the flow stresses resulting from the FEA are plotted in this graph to verify that they lie on the ML yield locus, as expected for ideal plasticity.

With these numerical examples, the applicability of the ML yield function developed in this work has been demonstrated for two fundamentally different kinds of flow behavior: anisotropic Hill-like plasticity and isotropic Tresca plasticity. The new formulation has proven to be numerically stable. The numerical effort is somewhat higher than that for mathematical closed-form yield functions, because the calculation of the predictor step requires a higher effort in numerically evaluating the distance of a given point in stress space to the yield locus. However, in conjunction with the implementation of the ML yield function in a compiled computer code, FEA even for large engineering models seems to be feasible with the new ML yield functions.

## 4. Discussion

In this section, the requirements on training data for the ML yield function will be more closely examined. For the first application of the ML yield function, training data for $n_{\mathrm{ang}}=36$ load cases have been constructed from a reference material with Hill-like anisotropy, and it has been demonstrated that this number of load cases produces rather accurate results in this case, whereas the approximation of a Tresca yield function required $n_{\mathrm{ang}}=600$ load cases. Numerical studies with larger data sets reveal that for Hill-like anisotropic yield functions the accuracy of the results increases slightly for data sets of up to $n_{\mathrm{ang}}=200$. The accuracy as well as the numerical stability and efficiency of the method remain stable for even larger data sets, which has been tested for up to $n_{\mathrm{ang}}=600$. It is also interesting to see that the accuracy of the method for Hill-like anisotropic yield functions is only slightly reduced when rather small data sets of $n_{\mathrm{ang}}=8$ load cases are used. Even producing the training data under plane-stress conditions with $\sigma_{3}=0$ does not change the quality of the results significantly, which is a consequence of mapping all stresses onto the deviatoric plane and extending the results by assuming that the material’s flow behavior does not depend on hydrostatic stress components, which is fulfilled to a very good degree for metals. If it is assumed, furthermore, that the material under consideration shows a symmetric flow behavior under tension and compression, only one half-space of the deviatoric plane needs to be characterized, and the results can be mirrored into the other half-plane. Thus, only the four load cases given in Table 2, have been sufficient to produce training data from which a useful ML yield function for Hill-like anisotropic plasticity results. This example is also provided in the Appendix A.

From these considerations, it can be concluded that for material with an anisotropic flow behavior that can be described with a Hill-like formulation, a small number of experiments under plane-stress conditions, as they can be performed on a bi-axial test rig, are fully sufficient to produce enough raw data to train the ML yield function. Of course, this experimental data could also be used to calculate the Hill parameters. However, the ML yield function offers a larger flexibility, and the training process for machine learning methods like support vector classification (SVC) is highly efficient.

Materials with a more irregularly shaped yield function, like the Tresca yield function, pose much higher demands on the available data and, furthermore, the parameters for the training of the SVC algorithm need to be adapted to allow for more flexibility. The obtained yield function still rounds off the sharp corners of the Tresca yield function such that the result resembles yield functions that can also be obtained with the 18-parameter model of Barlat et al. [4]. Again, it would be possible to determine these parameters directly from the available data, however, with an even higher effort than for the Hill parameters. Yet, the training effort for the ML yield function remains the same, independent of the volume of training data.

In general, it has been verified that only knowing the full stress tensors at the onset of plastic yielding is sufficient to train the ML yield function, and further information, e.g., on plastic strains, is not required. The only relevant information that needs to be known is whether for a given stress state the material response is purely elastic or elastic-plastic. Concerning the best strategy to produce training data, the raw data should lie in close proximity to the yield surface to cover the onset of yielding in an accurate way. Since support vectors are only produced from training data closest to the hyperplane separating the two categories “elastic” and “plastic”, however, using only this raw data would cause the SVC decision function (Equation 18) to drop to zero quickly outside the region covered by the support vectors. Hence, it is important to downscale the “elastic” raw data points and to upscale the “plastic” raw data points to finally cover the entire relevant region of the deviatoric stress plane and to produce sufficiently many support vectors in this domain. Comparing Figure 3 (Hill-like yield function) and Figure 8 (Tresca yield function), the influence of the number of training points is quite clearly seen, as in the former case 145 support vectors are created during the training process, whereas in the latter case 1689 support vectors are necessary to describe the sharp corners of the Tresca yield function. This example also demonstrates how the number of 36 vs. 600 raw data points is reflected in the number of generated support vectors.

Data-oriented constitutive modeling thus requires only a rather limited amount of data, as compared to other approaches in data-driven mechanics [9,11,13]. Large data volumes, of course, help to increase the accuracy of the resulting ML yield function, but the results achieved even with small amounts of data provide already a very good estimate of the material’s anisotropic flow behavior even for load cases that have not been tested. These comparatively moderate requirements on the volume of training data are a consequence of exploiting physical symmetry conditions on the material’s flow behavior in the formulation of the data-oriented yield function. For materials exhibiting a significant influence of the hydrostatic stress component on the plastic behavior, the method is still applicable, but the requirements on the training data will be higher.

Another aspect to be discussed here is the use of micromechanical models to produce training data. With such models, the mechanical behavior of realistic microstructures can be simulated with crystal plasticity methods [7], providing an accurate description of the plastic properties of polycrystaline metals with different microstructures and crystallographic textures. One disadvantage of such micromechanical models is their tremendously high numerical effort making them unsuited for FEA applications of engineering structures that are much larger than the grain size of a material. However, it is possible to employ relatively small micromechanical models, validated by experimental data, for creating a sufficiently large data volume describing the mechanical properties of the real material under various loading conditions with a high accuracy. By purposefully varying the microstructure or the texture of the model material, micromechanical simulations also serve the purpose of extending experimental data. This hybrid experimental and numerical data can then be used for the training of the ML yield function presented in this work. In this way, material parameters like grain size, grain morphology, and crystallographic texture can be explicitly included into the feature vector of the ML yield functions, in addition to the purely mechanical data used currently as input for the yield function. This microstructure-sensitive ML yield function can then be used in large-scale FEA for the simulation of engineering structures, which holds the possibility to consider the trained ML flow rule as a “digital twin” of the material, containing all relevant information on the material properties. The data-oriented constitutive model developed in this work will, hence, also pave the way for new approaches to scale-bridging materials modeling.

Critical issues that remain to be solved before the new method can be applied generally in FEA, include (i) the augmentation of the formulation with respect to shear components of stress and strain, (ii) regularization of the ML yield function to ensure its convexity and to the reduce noise in its gradient, and (iii) a data-oriented formulation of work-hardening. Concerning the latter point, it is noted that the current formulation allows the use of the standard methods of isotropic and kinematic work hardening if the hardening parameters are known, because it already contains the gradient of the yield function to calculate the plastic strain rate and also the tangent stiffness. However, a data-oriented formulation of work hardening, e.g., following the ideas of Chinesta et al. [13], would be more consistent with the idea of data-oriented constitutive modeling.

## 5. Conclusions

In this work, a new formulation of a data-oriented constitutive model for plasticity has been derived and applied within finite element analysis. The central element of this new constitutive model is a support vector characterization (SVC) algorithm serving as yield function. This SVC algorithm is trained by using deviatoric stresses as input data and the information whether a given stress state leads to purely elastic or rather to elastic-plastic deformation of the material as result data. In this way, a machine learning (ML) yield function is obtained, which can determine whether a given stress state lies inside or outside of the elastic regime of the material. Furthermore, the yield locus, i.e., the hyperplane in stress space on which plastic deformation occurs, can be reconstructed from the SVC, and the gradient on this yield locus can be conveniently calculated. Therefore, the standard formulations of continuum plasticity, as the return mapping algorithm, can be applied in finite element analysis in the usual way. Thus, it has been demonstrated that the new ML yield function can replace conventional yield functions in finite element analysis. The main advantage of such data-oriented constitutive models over the conventional ones is that they can be used with higher-dimensional feature vectors combining mechanical stresses with microstructural parameters of a material. In forthcoming work, it will thus be demonstrated how a single ML yield function can be trained to be used as a constitutive rule for a material in different microstructural states. The production of training data by micromechanical models, based on crystal plasticity and a discrete representation of the material’s microstructure, allows the ML flow rule to serve as efficient homogenization scheme, which offers new possibilities in scale-bridging material modeling.

## Figures and Tables

**Figure 1 materials-13-01600-f001:**
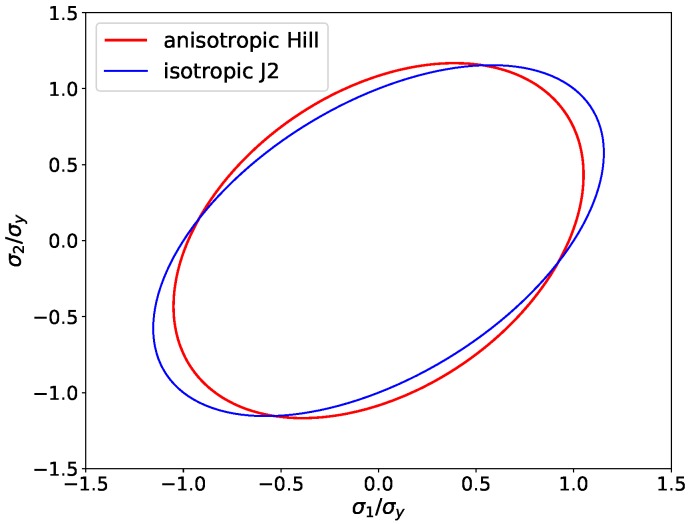
Yield locus for plane-stress conditions ($\sigma_{3}=0$) and Hill-like anisotropy with parameters given in Table 1 (red line) and for an isotropic material with the same yield strength, but $H_{1}=H_{2}=H_{3}=1$ (blue line). The values of the principal stresses are normalized by the yield strength $\sigma_{y}$.

**Figure 2 materials-13-01600-f002:**
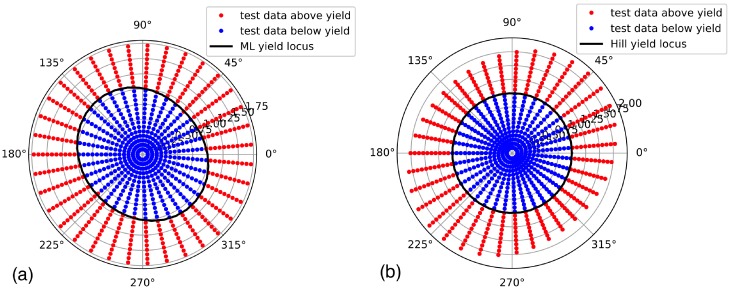
Polar plots of a subset of the training data produced from the anisotropic yield function of the reference material: (**a**) Von Mises (J2) equivalent stresses according to Equation (Equation 3) are used, such that the yield strength, rather than the equivalent stress, is a function of the polar angle θ. (**b**) Equivalent stresses are calculated according to the Hill definition in Equation (Equation 5) to achieve a constant yield strength by mapping the equivalent stresses accordingly. In both figures, the yield locus is indicated by a solid black line, data points in the elastic regime are plotted in blue color and data in the plastic regime in red color. Both figures represent the same stress data, only mapped in a different way; all stresses are normalized with the reference yield strength $\sigma_{y}$.

**Figure 3 materials-13-01600-f003:**
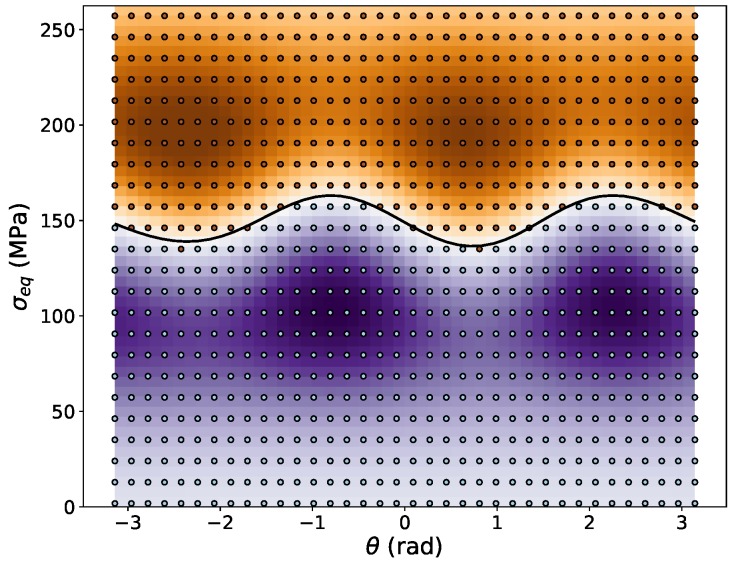
Field plot of the trained SVM decision function defined in Equation (Equation 18), where areas in purple color shades represent negative values and brown shades represent positive values. The numerical value of the decision function is not relevant because only its sign is taken into account in the flow rule. The isoline for $f_{\mathrm{SVC}}=0$ is represented as a black line. Training data are plotted in light blue color for data with negative values (elastic) and in brown color for positive values (plastic).

**Figure 4 materials-13-01600-f004:**
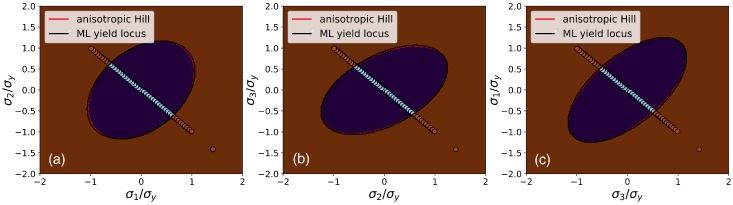
Color map of the trained SVC prediction of the yield function in slices through the principal stress space defined by plane-stress conditions: (**a**) $\sigma_{3}=0$; (**b**) $\sigma_{1}=0$; (**c**) $\sigma_{2}=0$. Brown regions indicate values of “+1” (plasticity) and purple regions values of “−1” (elasticity). The ML yield locus, corresponding to the isoline for $f_{\mathrm{SVC}}=0$, is represented as a black line; the yield locus of the Hill-like anisotropic reference material is indicated as a red line. The training data points are plotted with the same color code as in Figure 3.

**Figure 5 materials-13-01600-f005:**
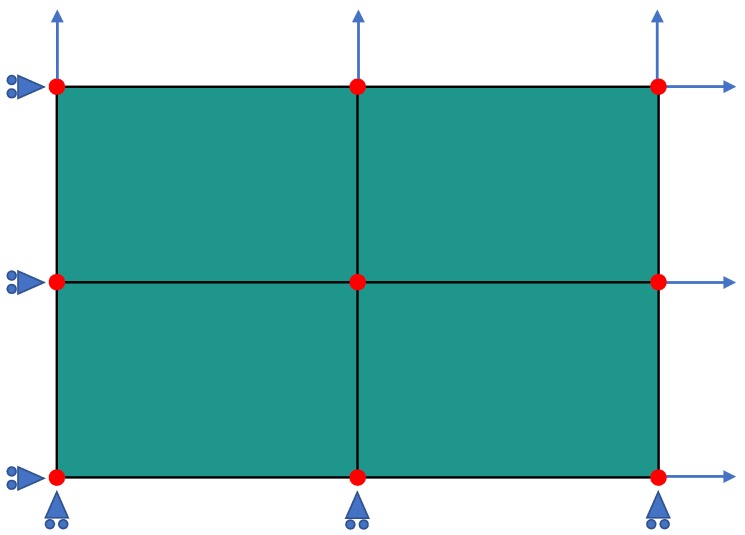
The finite element model on which four different load cases are studied consists of four quadrilateral elements (green) with linear shape function, and at total of nine nodes (red) situated at the corners of the elements. The bottom and left-hand-side boundary nodes are restricted to zero normal displacement (blue triangles), and the loading is applied on top and right-hand-side-nodes (blue arrows), as described in the text.

**Figure 6 materials-13-01600-f006:**
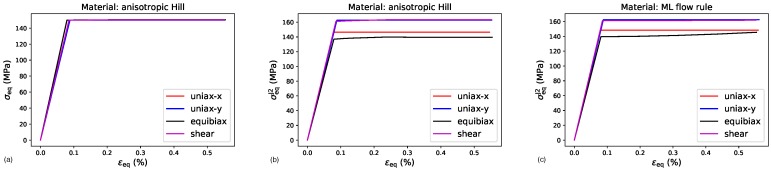
Stress strain curves obtained for elastic-ideal plastic material behavior under the loading conditions specified in the legend: (**a**) Equivalent total strain vs. equivalent Hill-stress, (**b**) equivalent total strain vs. equivalent J2-stress for Hill-like yield function, and (**c**) equivalent total strain vs. equivalent J2-stress for ML yield function.

**Figure 7 materials-13-01600-f007:**
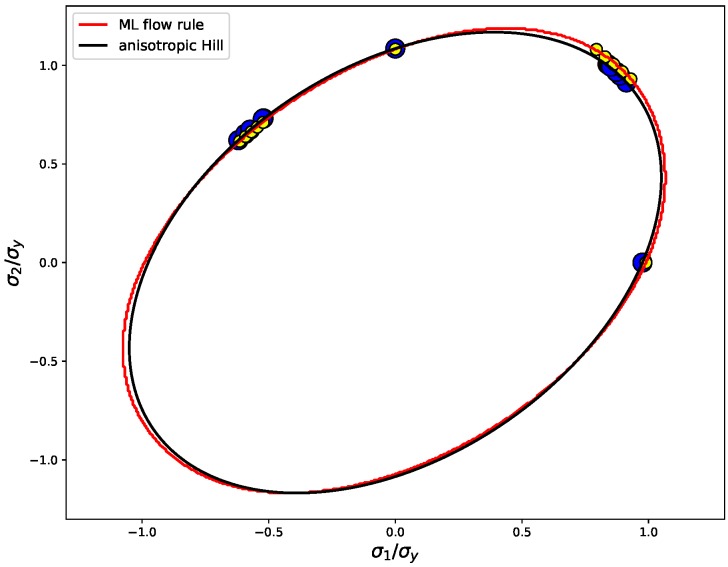
Stress states obtained for the four different plane-stress load cases are plotted in the $\sigma_{1}$-$\sigma_{2}$ plane together with the yield loci of the trained ML flow rule and the Hill-like reference material. The flow stresses resulting from the ML yield function are plotted as small yellow circles and those from anisotropic Hill plasticity as large blue circles.

**Figure 8 materials-13-01600-f008:**
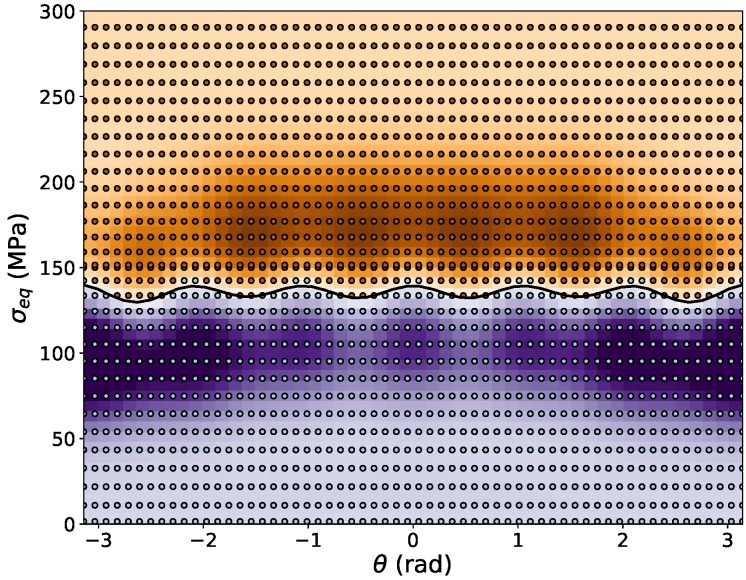
Field plot of the ML yield function trained with data generated from a reference material with a Tresca yield criterion, where areas in purple color shades represent negative values and brown shades represent positive values. The isoline, where the ML yield function is zero, is plotted as black line. The test data points are plotted as brown circles, for stresses in the plastic regime, and a s light blue circles for stresses in the elastic regime.

**Figure 9 materials-13-01600-f009:**
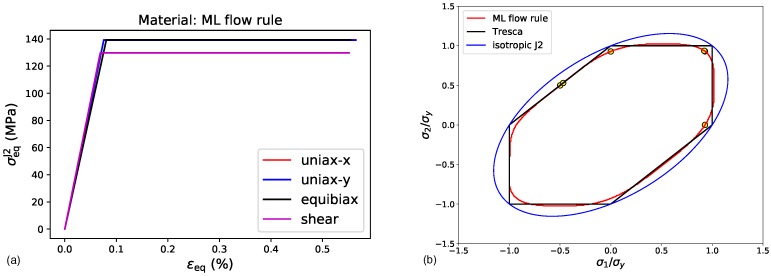
Results of FEA on the material with an ML flow rule trained with data obtained from a Tresca yield criterion: (**a**) Equivalent J2 stress plotted over the equivalent total strain, for the four different load cases given in Table 2. (**b**) Flow stresses obtained with the ML yield function plotted as yellow circles in the $\sigma_{1}$-$\sigma_{2}$ principle stress space, together with the yield loci of the ML flow rule, the Tresca flow rule and an isotropic J2 flow rule.

**Table 1 materials-13-01600-t001:** Elastic and plastic material parameters defining the reference material with Hill-like anisotropy in its plastic flow behavior. For simplicity, isotropic elastic behavior and ideal plasticity without work hardening are assumed in this work.

Quantity	Symbol	Value
Young’s modulus	*E*	200 GPa
Poisson’s number	ν	0.3
Yield strength	$\sigma_{y}$	150 MPa
Hill parameters	$H_{1},H_{2},H_{3}$	0.7, 1, 1.4

**Table 2 materials-13-01600-t002:** Yield stresses (YS) obtained for Hill-like yield function, with parameters given in Table 1, and machine learning (ML) yield function under the specified load cases. The relative errors in yield stress and equivalent plastic strain (PE) at maximum load are also specified.

Load Case	YS-Hill (MPa)	YS-ML (MPa)	Rel. Error YS	Rel. Error PE
uniaxial stress, horizontal	146.4	148.4	1.41%	−1.95%
uniaxial stress, vertical	162.7	162.2	−0.3%	−0.45%
equibiaxial strain	136.9	139.5	1.88%	−1.36%
pure shear strain	161.1	159.7	−0.87%	0.24%

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
