# Peer review of "Data-Oriented Constitutive Modeling of Plasticity in Metals"

_materials, 2020, doi:10.3390/ma13071600_

Round 1

Reviewer 1 Report

The paper presents application of ML algorithm to define the anisotropic yield surface of metallic materials and verifies possibility to apply it in FE element analysis. The subject is on the frontier of current research in mechanics, however there are some important inconsistencies in terms of model formulation that need to be corrected before paper can be accepted for publications.

My fundamental concern is related to the anisotropy treatment in the paper. The form of the yield condition given by definition (5) of equivalent stress is not correct. When analyzing anisotropic materials one must specify not only principal stresses but also principal directions with respect to anisotropy axes. Is author considering only the class of stresses for which these two sets of axes coincide? If so, then the solution is not general. The proposal that "In this orientation the stress tensor becomes a diagonal tensor, and Eq. (5) can be evaluated with parameters HI' in the rotated state of the material axis" is not an option - H_i are not then material parameters. How does author imagine to use it in FE calculations when the stress states can have different principal directions at each point of the analysis? This also applies when one uses polar coordinates as in section 2.2. In summary, you cannot correctly formulate the yield condition for anisotropic (even orthotropic!) material using only principal stresses without specifying what is the relation between principal directions and main axes of orthotropy. In Eq. (10) author specifies the gradient of the yield surface but he doesn't specify the plastic strain rate. It seems that in this case he assumes that the plastic strain rate will have the same principal directions as stress, which is not always the case for orthotropic materials. This basic issue of anisotropy description must be re-thought in the revision of the paper.

Other remarks:

  1. Introduction - it is not clear in what aspects the presented approach differs from the other machine learning proposals cited in the paper.
  2. Page 4, lines 93-95. The statement "whereas in his original work, Hill introduced three more parameters to scale also the shear components of the stress. Hence, the formulation introduced here, will apply for materials with orthotropic flow anisotropy". is not true. The original Hill condition with 6 parameters is restricted to orthotropy. To correctly describe orthotropic material one needs all six parameters. Note that Hill assumes that sigma_ij in his yield function are stress components in main axes of orthotropy
  3. Page 6, lines 118-119 "Furthermore, for symmetry reasons, it is required that the yield strength is a periodic function of the polar angle." What is the period?
  4. Page 6, lines 122-124. Statement "For isotropic plasticity, Dfd/Dtheta = 0, and in this case it is particularly easy to calculate the gradient and to see that the formulations in both coordinate systems result in the same gradient" is not precise. This is only the case for Huber-VonMises isotropic plasticity. For other isotropic yield conditions the function may depend on theta - like e.g. for the Tresca condition or the Hosford condition.
  5. Page 6, last paragraph: Sentence: "Data points that lie
    deeper within the elastic and plastic regions are required to prevent the decision function from falling back to zero, which would produce erroneous results." is not clear to me.
  6. Page 7, lines 140-141: Where is C in (18) or (19)?
  7. Author has used relatively simple case of yield surface to prove validity of the training procedure. Note that the example is quasi-linear since the yield surface is quadratic in stresses and its gradient easy to calculate. How the procedure would behave if one uses more advanced surfaces - like Barlat-Lian surface mentioned in the introduction and later in Discussion, which is not an ellipsoid but the modified Tresca polyhedron with rounded-off corners?
  8. Fig. 4. This is not clear to me why the training points produce straight lines in all three figures. It would mean that the ratio of principal stress e.g. s_1 and s_2 is constant for all the points?
  9. Section 3.2 It is not clear what FE analyses are done in four cases. Is this one-element analyses? How the boundary conditions are imposed for these four loading cases?
  10. Why 2/3 is used in definition (25)? It has sense only when epsilon is deviatoric part of strain not a total one.

Author Response

The author thanks the reviewer for the in-depth analysis of the manuscript and the constructive criticism that helped to significantly improve the paper and to increase its usefulness for the intended readership. In the following, all questions raised by the reviewer are addressed and the according changes in the manuscript are typeset in red text color.

Comment:

The paper presents application of ML algorithm to define the anisotropic yield surface of metallic materials and verifies possibility to apply it in FE element analysis. The subject is on the frontier of current research in mechanics, however there are some important inconsistencies in terms of model formulation that need to be corrected before paper can be accepted for publications.

My fundamental concern is related to the anisotropy treatment in the paper. The form of the yield condition given by definition (5) of equivalent stress is not correct. When analyzing anisotropic materials one must specify not only principal stresses but also principal directions with respect to anisotropy axes. Is author considering only the class of stresses for which these two sets of axes coincide? If so, then the solution is not general. The proposal that "In this orientation the stress tensor becomes a diagonal tensor, and Eq. (5) can be evaluated with parameters HI' in the rotated state of the material axis" is not an option - H_i are not then material parameters. How does author imagine to use it in FE calculations when the stress states can have different principal directions at each point of the analysis? This also applies when one uses polar coordinates as in section 2.2. In summary, you cannot correctly formulate the yield condition for anisotropic (even orthotropic!) material using only principal stresses without specifying what is the relation between principal directions and main axes of orthotropy.

Response: The reviewer is right in their observation that in this work, it is inherently assumed that the material is only loaded under purely principal stresses and that the material axes are aligned with the principal stress axes. This is now detailed in lines 97ff in the revised manuscript.

Otherwise, it is not uncommon in FEA to apply local coordinate systems at any Gauss point and to rotate the material axes into this coordinate system, e.g. in crystal elasticity and crystal plasticity, it is a standard procedure to rotate the elastic constants and the orientations of the slip systems into a local system at each Gauss point. Hence, the suggested procedure is feasible in principle. However, since generalizing the formalism to this case is out of the scope of this initial paper, this issue is pointed out more explicitly and the claim of generality is restricted, until the possibility of rotation the material axes is demonstrated in forthcoming work, see new text in lines 106ff and 378. As indicated by the reviewer, this work is on the “frontier of current research in mechanics” and, hence, it cannot be expected that all problems are solved in the first approach. However, all assumptions and simplifications that have been made are now clearly stated in the text. In fact, as pointed out by the other reviewers, there is a number of issues that needs to be solved, before the method can be generally applied in FEA, e.g. work hardening and robustness of the training, cf. lines 377ff.

Comment:

In Eq. (10) author specifies the gradient of the yield surface but he doesn't specify the plastic strain rate. It seems that in this case he assumes that the plastic strain rate will have the same principal directions as stress, which is not always the case for orthotropic materials. This basic issue of anisotropy description must be re-thought in the revision of the paper.

Response: The plastic strain rate is given in Eq. (6), following the established Prandtl-Reuss flow rule. The new Fig. 7 is added to demonstrate that, as stated by the reviewer, the plastic strain rate does not follow the directions of the principal stress. In this new figure, it is seen that the plastic strain is evaluated consistently with the expectation that the stress remains on the yield locus, as required in ideal plasticity, see new text in lines 266ff.

Other remarks:

    Introduction - it is not clear in what aspects the presented approach differs from the other machine learning proposals cited in the paper.

Response: The novelty of this work is worked out in a more detailed way in the Introduction, see lines 58-62.

Remark:

    Page 4, lines 93-95. The statement "whereas in his original work, Hill introduced three more parameters to scale also the shear components of the stress. Hence, the formulation introduced here, will apply for materials with orthotropic flow anisotropy". is not true. The original Hill condition with 6 parameters is restricted to orthotropy. To correctly describe orthotropic material one needs all six parameters. Note that Hill assumes that sigma_ij in his yield function are stress components in main axes of orthotropy

Response: The author thanks the reviewer for pointing out this mistake in the original manuscript, which has been corrected in the revised version, see lines 97ff.

Remark:

    Page 6, lines 118-119 "Furthermore, for symmetry reasons, it is required that the yield strength is a periodic function of the polar angle." What is the period?

Response: The yield function needs to cover the entire space, such that the periodicity in the polar angle is 2 pi, which is made clear in the revised version, line 128.

Remark:

    Page 6, lines 122-124. Statement "For isotropic plasticity, Dfd/Dtheta = 0, and in this case it is particularly easy to calculate the gradient and to see that the formulations in both coordinate systems result in the same gradient" is not precise. This is only the case for Huber-VonMises isotropic plasticity. For other isotropic yield conditions the function may depend on theta - like e.g. for the Tresca condition or the Hosford condition.

Response: The statement has been clarified by making clear that it only holds within the formalism of this work, line 132.

Remark:

    Page 6, last paragraph: Sentence: "Data points that lie deeper within the elastic and plastic regions are required to prevent the decision function from falling back to zero, which would produce erroneous results." is not clear to me.

Response: In the revised version, it has been clarified that the SVC decision function drops to zero, when is out of the range covered by the support vectors. This must be avoided by enforcing more support vectors lying in the plastic regime, i.e. at large equivalent stresses, such that the SVC decision function remains at a positive value, lines 162ff.

Remark:

    Page 7, lines 140-141: Where is C in (18) or (19)?

Response: The parameter C is only needed during the training, but it is not part of the solution. This has been pointed out explicitly in the revised manuscript, line 149f.

Remark:

    Author has used relatively simple case of yield surface to prove validity of the training procedure. Note that the example is quasi-linear since the yield surface is quadratic in stresses and its gradient easy to calculate. How the procedure would behave if one uses more advanced surfaces - like Barlat-Lian surface mentioned in the introduction and later in Discussion, which is not an ellipsoid but the modified Tresca polyhedron with rounded-off corners?

Response: The author thanks the reviewer greatly for pointing this out. To demonstrate the applicability also in this case, another example has been added to the revised manuscript, in which the method is applied to a Tresca yield criterion as reference material, see subsection 3.3 (lines 274ff with Figs. 8 and 9).

Remark:

    Fig. 4. This is not clear to me why the training points produce straight lines in all three figures. It would mean that the ratio of principal stress e.g. s_1 and s_2 is constant for all the points?

Response: All training stresses lie on the deviatoric plane, such that only for purely deviatoric loading cases the ratio of the principal stresses would remain constant. This has been made clear in the revised manuscript. It is also noted that the only purpose of the training stresses is to provide information to the SVC algorithm which stresses produce purely elastic strain and which stresses resulting from elastic predictor steps end in the plastic region, such that the return mapping algorithm needs to be invoked. Hence, the choice of the training stresses is somewhat arbitrary, the only physical information lies in the decision whether any given stress lies in the elastic or in the plastic region, see lines 162ff.

Remark:

    Section 3.2 It is not clear what FE analyses are done in four cases. Is this one-element analyses? How the boundary conditions are imposed for these four loading cases?

Response: More information is provided on the FEA. The models consisted of 4 quadrilateral elements under plane stress condition, and the boundary conditions are given explicitly in the revised version of the paper, see the new Fig. 5 and the new text in lines 251ff.

Remark:

    Why 2/3 is used in definition (25)? It has sense only when epsilon is deviatoric part of strain not a total one.

Response: Thanks for pointing this out, since epsilon is the total strain, the definition has been corrected in Eq. (25) and the numerical data in Fig. 6 is updated.

Reviewer 2 Report

  The author proposes new data-oriented formulation of yield surface.  The author shows that the formulation efficiently uses machine learning algorithm and can be easily implemented to conventional FEM without losing generality.  The data-oriented formulation is compared with Hill-like formulation, which demonstrates the applicability of the author’s model.  The author repeatedly mention that the potential application of the author’s model is not limited to the presented example but include microstructural features in forthcoming work.  Broad potential application would be of great interest to the readers.  This paper is an important contribution and I recommend that it be accepted for publication.  I have only a few comments listed below:

  1. In discussion, the authors discuss the relationship between the number of the data sets and the accuracy of the model. In my understanding, decision function does not use all of the input data sets but uses only support vectors. I think discussion of the number of the support vectors or location of the support vectors would be of interest to the readers.
  2. The author emphasizes that the model can include the microstructural features in future study. But, in my opinion, some of the readers would be of interest to the work hardening. I think it would improve readers’ understanding if the author clearly mentions about applicability to the materials which undergo work hardening.

Author Response

The author thanks the reviewer for the positive evaluation of the work and the constructive comments. All modifications compared to the original manuscript are typeset in red textcolor in the revised version.

Comment 1:

    In discussion, the authors discuss the relationship between the number of the data sets and the accuracy of the model. In my understanding, decision function does not use all of the input data sets but uses only support vectors. I think discussion of the number of the support vectors or location of the support vectors would be of interest to the readers.

Response: The reviewer is right with the statement that not all training data influence the resulting the support vectors, and that the trained decision function only depends on the latter. The relations between the training data and the resulting support vectors, and, hence, the quality of the trained model are explained in more depth in the revised manuscript, see lines 162ff and 338ff.

Comment 2:

    The author emphasizes that the model can include the microstructural features in future study. But, in my opinion, some of the readers would be of interest to the work hardening. I think it would improve readers’ understanding if the author clearly mentions about applicability to the materials which undergo work hardening.

Response: The possibility of applying this method to work hardening has been addressed explicitly in the revised manuscript, see lines 377ff.

Reviewer 3 Report

I found the paper well written and interesting.

In my opinion, it can be published as it is.

Author Response

The author thanks the reviewer for the appreciation of the paper.

Round 2

Reviewer 1 Report

The Author has sufficiently amended the manuscript in view of my remark.

It can now be accepted for publication in Materials.